# Estimation of the Circadian Phase Difference in Weekend Sleep and Further Evidence for Our Failure to Sleep More on Weekends to Catch Up on Lost Sleep

**DOI:** 10.3390/clockssleep7040067

**Published:** 2025-11-27

**Authors:** Arcady A. Putilov, Evgeniy G. Verevkin, Dmitry S. Sveshnikov, Zarina V. Bakaeva, Elena B. Yakunina, Olga V. Mankaeva, Vladimir I. Torshin, Elena A. Trutneva, Michael M. Lapkin, Zhanna N. Lopatskaya, Roman O. Budkevich, Elena V. Budkevich, Marina P. Dyakovich, Olga G. Donskaya, Dmitry E. Shumov, Natalya V. Ligun, Alexandra N. Puchkova, Vladimir B. Dorokhov

**Affiliations:** 1Independent Research Group for Biomedical Systems Math-Modeling, 12489 Berlin, Germany; 2Department of Normal Physiology, Medical Institute of the Peoples’ Friendship University of Russia, 117198 Moscow, Russia; 3Department of Physiology, Ryazan State Medical University, 390026 Ryazan, Russia; 4Department of Physiology, Medical Institute of Surgut State University, 628403 Surgut, Russia; 5Laboratory of Nanobiotechnology and Biophysics, North-Caucasus Federal University, 355029 Stavropol, Russia; 6Department of Economics, Marketing and Personnel Management, Angarsk State Technical University, 665835 Angarsk, Russia; 7Laboratory of Sleep/Wake Neurobiology, Institute of Higher Nervous Activity and Neurophysiology of the Russian Academy of Sciences, 117865 Moscow, Russia

**Keywords:** morningness–eveningness, chronotype, sleep timing, circadian phase, sleep–wake regulation, two-process model, simulation

## Abstract

The circadian phase difference between morning and evening types is a fundamental aspect of chronotype. However, results of categorizations into chronotypes based on reported sleep times show low concordance with those based on measurements of the hormonal or physiological or molecular rhythm–markers of the circadian phase. This might be partially explained by the profound individual differences in the phase angle between the sleep–wake cycle and these rhythms that depends on chronotype, age, sex, and other factors. Here, we examined the possibility of using self-reported sleep times in the condition of 5-days-on/2-days-off school/work schedule to estimate circadian phase differences between various chronotypes. In an in silico study, we determined that, for such an estimation, similarities of the compared chronotypes in weekend sleep duration and weekend–weekday gap and in risetime are required. In the following empirical and simulation studies of sleep times reported by 4940 survey participants, we provided examples of the estimation of circadian differences between chronotypes, and the model-based simulations of sleep times in morning and evening types exemplified a way to confirm such estimations. The results of in silico, empirical, and simulation studies underscore the possibility of using bedtimes and risetimes for direct estimation of the circadian phase differences between individuals in real-life situations, such as a 5-days-on/2-days-off school/work schedule. Additionally, the results of these studies on different chronotypes provided further mathematical modeling and empirical evidence for our failure to sleep more on weekends to recover/compensate/pay back/ catch up on lost sleep.

## 1. Introduction

The terms “morningness–eveningness” [1] and “chronotype” [2] were coined to describe the natural tendencies of an individual to sleep, wake, and optimally perform during certain hours of the day. The underlying timing of the biological clock dictates later timing of sleep, wakefulness, and maximal productivity for evening types compared to morning types [3,4]. The reviews of the literature on individual variation in the domain of chronobiology [5,6,7,8] consider the circadian phase difference between morning and evening types as a fundamental aspect of chronotype.

Either their real [9,10,11,12] or preferred times for sleep and wakefulness are usually used to assign individuals to these distinct morning and evening types [13,14,15]. Despite the utilization of sleep and wake times for chronotyping, it is believed that these times cannot provide an accurate estimate of the difference between morning and evening types in the circadian phase, especially when these sleep times are reported in the typical condition of 5-days-on/2-days-off school/work schedule. Instead, the conventional hormonal and physiological markers of the circadian phase position were often used to measure this circadian phase difference between these types, such as the onset of melatonin secretion and the minimum of core body temperature [3,4,16,17,18,19,20]. More recently, there were also attempts to introduce molecular markers, such as the phases of expression of several circadian clock genes [21,22,23] and a gene expression pattern at a single time point [24].

However, it was also demonstrated that the correlations of the actual circadian phase of the melatonin rhythm with either actual or preferred sleep times are not strong [25,26], and that chronotype categorizations based on assessments of sleep times demonstrate low concordance with those based on a measurement of the circadian melatonin phase [26]. Since the sleep–wake cycle and melatonin rhythm are two very different circadian rhythms, one of the reasons for such a low concordance between these two methods of chronotyping can be the differences in phase angle between these circadian rhythms. This phase angle is known to vary from person to person depending on chronotype, age, sex, and other factors (e.g., [3,4]).

Consequently, we examined the possibility of direct estimation of the circadian phase difference from sleep times self-reported in the condition of a 5-days-on/2-days-off school/work schedule, using, as an example, such a difference between various chronotypes. In the present in silico study, we applied one of the versions of the two-process model of sleep–wake regulation to determine the particular relationships between sleep times reported for weekdays and weekends, allowing the direct estimation of the difference between chronotypes in the circadian phase of sleep. In the following analysis of an empirical dataset, we used sleep times reported by almost 5000 survey participants to identify these particular relationships between sleep times and to exemplify a proposed methodology for such a direct estimation. Finally, we simulated sleep times reported by morning and evening types to exemplify a way of confirming the estimated circadian phase difference in terms of the parameters of the sleep–wake regulating process. We hypothesized that the results of the present in silico, empirical, and simulation studies can provide evidence for the possibility of estimating circadian phase differences between individuals from sleep times reported in real-life situations, such as a 5-days-on/2-days-off school/work schedule.

For the last four decades, a quantitative version of the two-process model of sleep–wake regulation [27] remains the major source of our understanding of the basic mechanisms generating the sleep–wake cycles. Our previous model-based simulations of weekday and weekend sleep suggested that daytime workers/learners are not “accumulating sleep debt” on weekdays to “pay it back” during the following weekends [28,29]. However, ironically, the concept of the homeostatic regulation of sleep duration proposed by the two-process model of sleep–wake regulation [27] implicitly inspired several explanations of the sleep–wake regulation in daytime workers/learners that were not supported by simulations based on this model [29]. As a result, a huge number of publications describe weekend sleep under misleading terms such as “recovery sleep” (e.g., [30,31,32]), “compensatory sleep” (e.g., [33,34,35]), “catch-up sleep” (e.g., [36,37,38]), and so on. Consequently, a failure to sleep more on weekends to recover/compensate/pay back/catch up on lost sleep was additionally supported here by the results of model-based prediction and comparison of sleep times reported by different chronotypes for weekdays and weekends.

## 2. Results

### 2.1. Results of the In Silico Study

In the in silico study, we used the same set of model parameters (Table 1, bottom part) to calculate three pairs of time courses of sleep and wakefulness on weekdays and weekends characterized by the identical weekend risetimes (Figure 1 and Table 1, upper part), weekday risetimes (Figure 2), and gaps between weekend and weekday risetimes (Figure 3). The model predicts that, irrespective of the gap between weekend and weekday risetimes (4.0 h and 2.0 h in Figure 1 and Figure 2), the same timing and duration of sleep are predicted to be restored during weekdays (Table 1, upper part). In other words, the prediction is that the endogenously determined sleep duration must be fully reestablished during two days of the weekend after any shift in weekday risetime relative to weekend risetime (e.g., either 4.0 h or 2.0 h in Figure 1 and Figure 2). As illustrated in Figure 1 and Figure 2, the model does not predict that the wake phase of the sleep–wake cycle on weekdays can be delayed after a shorter sleep duration on the previous weekdays. The figures show that the wake phase of the sleep–wake cycle on weekdays is always terminated at the upper threshold of deviation from the setpoint allowed by the sleep–wake regulator to initiate the transition to the sleep phase instead of rising further to accumulate “sleep debt” (i.e., see the upper sine function in Figure 1 and Figure 2). This implies that a shorter weekday sleep duration (i.e., a larger weekday sleep loss or, in other terms, a larger “accumulated sleep debt”) cannot cause a larger weekend sleep duration (i.e., in other terms, “compensation of sleep loss” or “paying sleep debt back” or “catching up on lost sleep”, etc.).

Moreover, the model also predicts that since a shorter weekday sleep duration cannot cause a larger weekend sleep duration (Figure 1), the difference between earlier and later weekday risers in weekend sleep timing (2.0 h in Figure 1 and Figure 2) is equal to the difference in the circadian phase of sleep (2.0 h in Figure 1 and Figure 2). If this is true, the difference between evening and morning types in the circadian phase of sleep can be estimated as the difference in weekend risetime (or in weekend bedtime). However, these in silico calculations do not take into account that a larger shift in weekday risetime relative to weekend risetime in later risers compared with earlier risers (4.0 h vs. 2.0 h in Figure 1 and Figure 2) can lead to a larger advancing shift in the 24 h cycle of their light exposure, which in turn can lead to a greater advance of the circadian clock phase, and consequently a larger advance of the wake and sleep phases of the sleep–wake cycle on weekdays and even on the following weekend in later risers. Therefore, a non-zero difference between earlier and later weekday risers in weekend risetime is expected instead of a zero difference shown in Figure 1 (i.e., an advance must be larger after an earlier weekday risetime due to a larger gap between weekend and weekday risetimes). Similarly, the calculations shown in Figure 2 do not take into account that such a larger shift in weekday risetime relative to weekend risetime (4.0 h vs. 2.0 h) can lead to a larger advancing shift in light exposure, and so on (see above). Therefore, the difference between earlier and later weekday risers in weekend risetime is expected to be smaller than their difference in the circadian phase (e.g., somewhat smaller than the 2.0 h difference shown in Figure 2).

The question arises: what might be a condition when this advancing shift in the 24 h cycle of light exposure is identical in earlier and later risers? The answer is: this is a condition when, in addition to the identity of durations of weekend sleep, the weekend–weekday gap in risetime is also identical in these risers. In contrast to the results of calculations shown in Figure 1 and Figure 2, the calculations shown in Figure 3 illustrate this condition of the identical shifts in weekday risetimes relative to weekend risetimes (2.0 h vs. 2.0 h in Figure 3 instead of 4.0 h vs. 2.0 h in Figure 1 and Figure 2). Therefore, when the advancing shifts in the 24 h cycle of light exposure are expected to be identical in earlier and later risers, the difference between them in weekend sleep times (either bed- or risetime) is equal to the difference between them in the circadian phase of sleep. Consequently, the difference between chronotypes in the circadian phase of sleep can be calculated from the sleep times reported for weekdays and weekends as the difference in weekend sleep timing in the condition of similarity of weekend–weekday gaps in risetime and weekend sleep duration (Figure 3).

Thus, the results of the in silico study indicated that sleep times can be used for direct estimations of the differences in the circadian sleep phase between chronotypes with similar weekend sleep duration and similar weekend–weekday gap in risetime (Figure 3). Therefore, data from the empirical study can be used to exemplify the application of this approach to the calculation of the difference between chronotypes in the circadian phase of sleep as the difference between them in weekend risetime (Table 2 and Appendix A).

Moreover, data from an empirical study can be used to perform the estimations of circadian phase difference using the whole dataset, i.e., individuals with different weekday gaps in risetime. The calculations shown in the notes to Table 3 suggest that the difference in circadian phase of sleep (Table 3, lower part) can be directly calculated as the sum of the difference between earlier and late weekday risers (Table 3, middle part) and the difference between morning and evening types (Table 3, upper part). This implies that the whole set of data on two compared chronotypes can be used to calculate a difference between these chronotypes in the circadian phase of sleep (Table 3, Appendix A, and Figure 4B).

### 2.2. Results of the Empirical Study (Sleep Times, Sleepiness, Sleepability, and Wakeability in Various Chronotypes)

The mean ages ± standard deviations in subsamples of 1271 male and 3075 female university students (age 17–25 years) were 19.3 ± 2.1 and 19.4 ± 1.7 years, respectively, and the mean ages ± standard deviations in subsamples of 138 male and 456 female lecturers and other staff members (age > 25 years) were 41.0 ± 13.7 and 42.5 ± 12.2 years, respectively.

As shown in Appendix A, the survey participants from two age subsamples were further divided into earlier and later weekday risers, based on whether or not their risetime was earlier or later than 7:00, and into seven chronotypes according to their answer to the SIC (Single-Item Chronotyping) question: “Self-assess your chronotype by choosing one of six patterns of daily change in alertness level”. The chronotypes are abbreviated as LIVEMAN types: Lethargic, Inconclusive, Vigilant, Evening, Morning, Afternoon, and Napping. The significant differences between these chronotypes were confirmed by assessing chronobiological characteristics such as weekend sleep times (Appendix A), and sleepiness, sleepability, and wakeability in the morning, daytime and evening/night hours (Appendix A).

### 2.3. Results of the Empirical Study (Difference in Circadian Phase of Sleep Between Various Chronotypes)

Appendix A contains the results of ANOVAs of sleep times in the subdivisions of the whole sample applied for testing the significance of four independent factors (seven chronotypes, two ages, two sexes, and two weekday risetimes). Appendix A contain the estimates of mean sleep times in the whole sample and two age subsamples (they are also illustrated in Appendix A). These ANOVAs (Appendix A) with the following post hoc pairwise comparisons (Appendix A) confirmed a model-based prediction of independence of weekend time in bed from the weekend–weekday gap in risetime (i.e., a shorter weekday time in bed caused by a larger gap does not lead to a longer weekend time in bed). Since this gap was significantly larger in evening than morning types, this excludes the possibility of simple comparison of the difference between all morning types and all evening types in the circadian phase of sleep using the difference in their weekend sleep timing (the latter difference is smaller due to a larger advance of the circadian phase in evening types due to a larger advance of the 24 h cycle of light exposure in response to early weekday wakeups).

Post hoc pairwise comparisons of weekend time in bed in chronotypes reported in Table 2 suggested that, despite the independence of this time from weekend–weekday gap in risetime predicted by the model (Figure 1), it was, nevertheless, shorter in one of six chronotypes (vigilant type) compared to lethargic, napping, and afternoon types. Indeed, such a shorter time in bed is expected for this vigilant type due to its chronobiological characteristics illustrated in Appendix A, e.g., low sleepiness levels, high wakeability, and low sleepability both in the morning and evening hours. Since weekend time in bed in the vigilant type was significantly different from that in some of the other types, this type was not included in the comparison of the circadian phase difference between all chronotypes presented in Appendix A.

The results suggesting non-significant differences in weekend time in bed and weekend–weekday gap in risetime in some of the chronotypes (e.g., evening vs. napping and lethargic vs. afternoon; Appendix A) provided the possibility to interpret the differences between these types in weekend risetime as estimates of the differences between them in their circadian phase of sleep (Appendix A). The results suggested that the circadian phase of the evening type was significantly delayed relative to the phase of the napping type, but this difference did not exceed one h (at least, 0.5 h), and the lethargic type did not significantly differ from the afternoon type in this phase (Appendix A).

Moreover, since weekend time in bed in morning types was not significantly different from time in bed in vigilant and other types, this provided the possibility to compare the circadian sleep phase of this type with circadian sleep phases of all the other types by means of separate ANOVAs of samples including only morning type and one of the other types (Table 3 and Appendix A and Figure 4 and Figure 5).

Figure 4A illustrates the similarity of chronotypes on the advancing shifts in weekend bed- and risetimes in response to the advancing shift in weekday risetime. This similarity is confirmed by non-significant interactions between the factors chronotype and weekday risetime (Appendix A, upper part). However, similar advancing shifts in morning and evening types also suggested that, irrespective of the shift, there was a profound difference between these types in the weekday–weekend gap in risetime (i.e., this weekend–weekday gap in risetime was larger in evening earlier weekday risers than in morning earlier weekday risers and it was also larger in evening later weekday risers than in morning later weekday risers; Appendix A). In contrast, this difference in the gap between these chronotypes was found to be non-significant in the comparison of earlier weekday risers of morning type with weekday risers of evening type (Appendix A, upper part). This non-significant difference provided the possibility to interpret the difference between these sub-subsamples of morning and evening types in weekend risetime as the estimate of the difference between them in the circadian phase of sleep (Appendix A). Such comparison of morning and evening types indicated that this difference exceeded two hours (Appendix A, upper part, Appendix A). The difference between earlier risers of morning type with later risers of three other types (lethargic, napping, and afternoon) was smaller by at least 0.5 h, i.e., the circadian phase position of lethargic, napping, and afternoon types was between the positions of morning and evening types. Similar estimates of circadian phase differences between morning and other chronotypes were obtained by using another approach to this estimation when ANOVAs were performed on datasets consisting of morning type and one of the other types (Table 3, Appendix A). Figure 4B illustrates the differences between chronotypes obtained by comparison of earlier risers of morning type (the earliest bed- and risetimes for this type) with later risers of evening, lethargic, napping, and afternoon types (the later bed- and risetimes for these types).

### 2.4. Results of the Empirical Study (Difference Between Ages in Rise- and Bedtimes)

The results illustrated in Figure 5A,B suggested that, as expected, the two age subsamples drastically differed in bed- and risetimes. These times on weekends in older age were advanced relative to the times in younger age (Appendix A, Figure 5). Despite this, two ages showed practically identical differences between chronotypes in the circadian phase of their sleep, estimated as the difference between early risers of morning type and later risers of evening type in weekend risetime (Figure 5, and Appendix A, upper parts, and Appendix A). Other compared pairs of chronotypes also demonstrated similarity of older and younger ages in the difference between chronotypes in weekend risetime (Appendix A, upper part, and Appendix A) despite the similarly large differences between ages in weekend sleep timing (Appendix A).

### 2.5. Results of the Simulation Study

Table 4 and Figure 6 exemplify the way in which a model-based simulation (Figure 6) can be applied to confirm the results of in silico (Figure 3) and empirical studies (Figure 4B). The simulation supported the results of direct estimation of the differences between chronotypes in the circadian phase of weekend sleep. In this example, the same set of model parameters, with the exception of the circadian phase parameter, was used to simulate rise- and bedtimes reported by earlier risers of morning type and later risers of evening type for weekdays and weekends. In particular, the results of this simulation (Table 4 and Figure 6) supported the results of the empirical study, indicating a more than 2.0 h difference between these two chronotypes in the circadian phase of sleep. They also supported the predicted (Figure 1, Figure 2 and Figure 3) and observed similarity (Table 2, Table 3, Appendix A) of weekend sleep duration in these two chronotypes.

## 3. Discussion

A fundamental aspect of chronotype is the circadian phase difference between morning and evening types. While preferred and actual times for sleep and wakefulness are usually used to distinguish between chronotypes, the circadian phase difference between them is additionally measured with the conventional hormonal, physiological, and molecular rhythms as markers of the circadian phase position. However, the literature suggests (e.g., [25,26]) that a result of categorizations into chronotypes based on sleep times can demonstrate low concordance with that based on the measurements of conventional rhythms as markers of the circadian clock, partly due to the variation in the phase angle between the sleep–wake cycle and these rhythm markers that depends on chronotype, age, sex, and other factors. In the present in silico study, we suggested a direct estimation of the differences between chronotypes in the circadian phase from sleep times reported for weekdays and weekends. In the following empirical and simulation studies, we provided examples of the estimation of the circadian difference between several chronotypes. Overall, these results of in silico, empirical, and simulation studies underscore the prospects of using self-reported sleep times for estimation of the circadian phase differences between individuals in real-life situations, such as a 5-days-on/2-days-off school/work schedule.

As expected, the difference in weekend sleep timing between subsamples of morning and evening types was found to be reduced due to a larger shift in the weekday circadian phase in evening types compared with morning types. When the subsamples of these types were further subdivided to provide comparison of participants with a similar weekend–weekday gap in risetime (with weekday risetime before 7:00 and later for morning and evening types, respectively), the difference between these types in weekend sleep timing exceeded two hours, which is in full agreement with the differences obtained in many studies of the circadian phase of hormonal, physiological, or molecular rhythm markers of the circadian clock in morning and evening types [3,4,16,17,18,19,20,21,22,23]. However, to validate the proposed method of calculation of the circadian phase difference from sleep times, future studies are required. For instance, the difference between morning and evening types in the circadian phase of sleep and the difference in the circadian phase of a rhythm marker of the circadian clock can be measured in the same study participants.

Additionally, we examined, for the first time, whether the circadian phases of morning and evening types differ from the phases of other chronotypes. As expected, the phases of these other types were found to be delayed relative to the phase of the morning type and advanced relative to the phase of the evening type. It has to be noted that, when only sleep timing (the sleep phase of the sleep–wake cycle) is used to classify individuals into chronotypes, such “bedroom typology” can be simple and unidimensional. However, the typology usually became more complex and multidimensional when, in addition to individual variation in sleep timing, it accounts for individual variation in the timing of activity and productivity (i.e., during the wake phase of the cycle). Therefore, research in the field of differential chronobiology shows the tendency to extend the term “chronotype” to other than morning and evening types [39,40,41,42]. In accord with such a wider conceptualization of the terms “chronotype” and “circadian type”, we applied the SIC [42] designed for multidimensional assessment of individual variations in the sleep–wake pattern. Future research can also be aimed at further elaboration of the proposed methodology to allow the estimation of the differences in the circadian phase of sleep between chronotypes characterized by significantly different time in bed on weekends (i.e., vigilant type vs. lethargic, napping, and afternoon type).

The results of in silico comparison of weekday and weekend times in bed in different chronotypes provided further mathematical modeling evidence for our failure to sleep more on weekends to recover/compensate/pay back/catch up on lost sleep. This evidence was confirmed by the results of the analysis of empirical data and their simulation. These results of comparison of sleep times in various chronotypes corroborate the previously published results of comparison of sleep times reported before and during lockdown, for early and later school start time, and for earlier and later weekday risetime [28,29].

The results on the circadian phase of sleep in morning and evening types have practical implications. Despite complete freedom to sleep in and nap during the two weekend days, evening types cannot change their endogenously determined late circadian phase, and, as suggested by the study results, they cannot reverse the reduction in sleep during the week by the extension of weekend sleep beyond its normal, adequate duration. Negative impacts of the disparity between chronotypes in weekday sleep loss underlie the complaints about “the tyranny of the early birds” (e.g., [43]). This disparity rests upon the tradition of setting working and school start times too early. Work/study culture is biased towards the circadian clocks of morning types, and, therefore, evening types are forced to sacrifice a larger amount of sleep on weekdays to arrive at their place of work/study at the morning type–oriented start times. We previously showed that the difference between earlier and later weekday risers is equal to the sum of the differences between these risers in weekday sleep loss and weekend sleep advance that can be measured as the differences in weekend risetime and weekend–weekday gap in risetime, respectively [29,44]. This equation highlights the possibility of reducing its first term (relative weekday sleep loss) by increasing the contribution of the second term (relative advance of weekend sleep timing). Therefore, one of the interventions aimed at reducing in disparity between evening and morning types in weekday sleep loss might be earlier morning exposure to higher-intensity light and an earlier termination of such an exposure in the evening. To achieve the goal of change in the ratio between the two terms of the equation, the previously approved light interventions (e.g., [45,46,47]) can be recommended for evening types.

Several limitations of this study require acknowledgement. The present results cannot be generalized across cultures, socioeconomic conditions, level of education, and the whole lifespan from the youngest to the oldest ages. Moreover, the results are based on self-reports of just four sleep times. They cannot be accurate, they were not compared with objectively measured sleep times, and they did not include the parallel measurements of phase positions of the molecular, physiological, or hormonal markers of the circadian clock. Moreover, we did not account for timing and weekend–weekday variability in light exposure. However, our major aim was to propose the methodology for direct estimations of the circadian phase difference from sleep times rather than to validate this methodology by obtaining accurate estimates of the difference between chronotypes in sleep times along with measurements of the phase of the conventional markers of the circadian clock.

## 4. Materials and Methods

### 4.1. Model-Based Computations (In Silico Study)

For the present in silico and simulation studies, we used one of the variants [48] of the two-process model of sleep–wake regulation [27] proposing that any parameter of the homeostatic process in the classical variant of the two-process model [27] is modulated by the circadian clock. This modulation [48] was included in the model as the counterpart of the circadian process in the classical variant of the model [27]. Mathematically, this circadian term is the simplest periodic (sine) function with the 24 h period (the period was assigned to 24.00 h because, in natural settings, the circadian clocks always remain under control of (i.e., are entrained to) the external light–dark cycle with the 24 h period [48]). If t1 and t2 are the initial times for the buildup and decay phases of this process of sleep–wake regulation $S\left(t\right)$, it can be calculated using the following equations:(1a)$$S\left(t\right)=[S_{u}+C(t)]-\{[X_{u}+C(t)]-S_{b}\;\}*e^{-(t-t1)/[Tb\;-\;k\;*\;C(t)]},$$
(1b)$$S\left(t\right)=[S_{l}+C(t)]-\{S_{d}-[S_{l}+C(t)]\}*e^{-(t-t2)/[Td-k\;*\;C(t)]}$$
where

(2)$$C(t)=A*sin(2\pi *t/24+\varphi_{0})$$
is a sine function with a 24 h period (Table 1 and Table 4).

The parameters of the model [48] were initially derived from data on (1) the durations of recovery sleep after 6 gradually increasing intervals of extended wakefulness [49] and (2) the levels of slow-wave activity (SWA) in 10 naps [50] and two recovery sleep episodes [51,52]. The simulation of such experimental data provided the possibility of measuring $S\left(t\right)$ as relative slow-wave activity (rSWA), i.e., rSWA = 1 in the baseline sleep. The initial parameters of the model were slightly modified in the previous (e.g., [28,29]) and present computations and simulations (Figure 1, Figure 2, Figure 3 and Figure 6, and Table 1 and Table 4) to account for the difference between the initially simulated experimental sleep durations [49] and times in bed calculated from self-reported bed- and risetimes on weekdays and weekends [28,29]. Detailed description of the model, its physical counterpart, and its parameters can be found in [28].

All calculations and simulations were performed using macros in Excel software. In the in silico study, the model of the sleep–wake regulating process [48] was used to calculate the sleep–wake cycles of two hypothetical earlier and later risers, and the results of the calculations are illustrated in Figure 1, Figure 2 and Figure 3. It was proposed that the process of sleep–wake regulation $S\left(t\right)$ is identical in these earlier and later risers (lower part of Table 1). It was additionally proposed that these risers have either identical weekend risetimes and circadian phases (Figure 1) or identical weekday risetimes but different circadian phases (Figure 2) or identical gaps between weekend and weekday risetimes but different circadian phases (Figure 3). The 24 h sleep–wake cycle in the model of this process $S\left(t\right)$ is represented by daily alternations of an inverse exponential buildup phase (1a) and an exponential decay phase (1b), i.e., the wake and sleep phases of the cycle. The parameters of $S\left(t\right)$ are additionally modulated by the circadian clock represented by a sine-form function with a 24 h period, $C\left(t\right)$ (2). In these figures, the modulation is illustrated by the 24 h oscillations of $S_{b}\left(t\right)\;$ and $S_{d}\left(t\right)$ that are the highest buildup and the lowest decay of $S\left(t\right)$, i.e., bedtime and risetime, respectively, determined by the endogenous sleep–wake regulating process $S\left(t\right)$ (1,2) measured in rSWA (see the model parameters in Table 1). These $S_{b}\left(t\right)$ and $S_{d}\left(t\right)$ are, in fact, the thresholds to which the sleep–wake regulator allows the deviations of the regulated process from the setpoint.

In Figure 1, the difference in risetime on weekends is equal to zero because, despite two different 2 h or 4 h gaps in weekend–weekday risetime, the circadian phases of sleep are identical (i.e., rise- and bedtimes on Sunday are equal to 9:00 and 24:00, respectively). The model predicts that, due to the circadian modulation $C\left(t\right)$ of the parameters of $S\left(t\right)$, the endogenously determined bed- and risetimes (shown for Sunday) are restored during just one day with ad lib sleep (between Friday and Saturday). This restoration is not affected by the advance of weekday risetime relative to weekend risetime (e.g., either 4.0 h or 2.0 h in earlier and later risers, respectively). Therefore, weekend rise- and bedtimes on Sunday are equal to 9:00 and 24:00, respectively, i.e., they are expected to be identical after 5 days of earlier and later weekday risetime at 5.0 a.m. and 7.0 a.m., respectively.

In Figure 2, not only are two different weekend–weekday gaps in risetime (2.0 h or 4.0 h) hypothesized, but also a 2.0 h difference in the circadian phase of sleep is hypothesized. This makes identical the weekday risetimes of earlier and later risers, and, therefore, the model predicts that, due to the combination of the 2.0 h difference in this phase with the identical 2.0 h difference in bed- and risetimes on weekends, the difference in weekday risetime is equal to zero.

In Figure 3, two identical (2.0 h) weekend–weekday gaps in risetime and the 2.0 h difference in the circadian phase of sleep are hypothesized. Since there is no difference in this gap, the difference in weekday risetime is equal to the difference in weekend risetime (and weekend bedtime either). Consequently, the results of model-based calculations shown in Figure 1, Figure 2 and Figure 3 suggest the following variants of the equation: difference in weekday risetime = difference in weekend risetime plus weekend–weekday gap in risetime. 1. The difference in weekday risetime is equal to the gap in risetime (Figure 1). 2. The difference in weekend risetime is equal to the gap in risetime (Figure 2). 3. The difference in weekday risetime is equal to the difference in weekend risetime (Figure 3).

The calculations shown in Figure 1 and Figure 2 do not take into account a larger advancing shift in the 24 h light exposure expected from a larger advancing shift in weekday risetime, while Figure 3 suggests that these shifts are expected to be identical because the difference in weekday risetime is equal to the difference in weekend risetime (i.e., the shifts in the 24 h light exposure are identical in earlier and later risers illustrated in Figure 3). Consequently, the difference in weekend sleep times is expected to be equal to the difference in the circadian phase of sleep due to a zero gap between weekend and weekday risetimes (Figure 3).

### 4.2. Survey Participants and Assessments of Chronotype (Empirical Study)

In the winter season, lecturers invited students and staff members of older age to respond to the questions asking about their sleep–wake habits and behavior (Appendix A). No exclusion criteria were applied for participation in the survey. All students attended classes/office in the university buildings, which excludes the possibility that they were in unhealthy physical and mental condition at the time of participation in the survey. The responses were collected from smartphones using the Google Forms questionnaire. Informed consent was obtained from each participant in the form of a response of “Agree” to the first statement of the questionnaire: “I give informed consent to anonymously and voluntarily participate in this online survey of sleep–wake behavior and habits”.

The survey participants were asked to report clock hours for bed- and risetimes on weekdays and weekends (Appendix A). Single-Item Chronotyping (SIC) [42] was used for the subdivision of the whole sample into chronotypes (Appendix A). Additionally, survey participants were asked to complete two multi-item chronotypological questionnaires (Appendix A): the 19-item Visuo-verbal Judgment Task (VJT) [53], for reporting expected levels of sleepiness on a 1.5-day interval of permanent wakefulness, and the reduced version of the 60-item Sleep–Wake Adaptability Test (SWAT) for self-assessment of abilities to sleep or wake on demand at different times of the day [54,55].

### 4.3. Details on Questionnaires for Assessment of Chronotype (Empirical Study)

The original English version of the SIC was proposed by O. Mairesse and A. Putilov for distinguishing between six rather than just two morning and evening chronotypes [42]. The English names for the seven response options to the SIC’s question (“Self-assess your chronotype by choosing one of six patterns of daily change in alertness level”) were abbreviated as “LIVEMAN” (Lethargic, Inconclusive, Vigilant, Evening, Morning, Afternoon, Napping type). The illustrations and short descriptions of levels of alertness in the morning, afternoon, and evening were assigned to the graphs and names of these chronotypes (see the SIC in the Appendix A to [42]): (1) Lethargic type (“Moderately energetic type” in Russian): low level in the morning, afternoon, and evening; (2) Vigilant type (“Highly energetic type” in Russian): high level in the morning, afternoon, and evening; (3) Evening type: low level in the morning, middle in the afternoon, high in the evening; (4) Morning type: high level in the morning, middle in the afternoon, low in the evening; (5) Afternoon type (“Daytime type” in Russian): low level in the morning, high in the afternoon, middle in the evening; (6) Napping type (“Daytime sleepy type” in Russian): high level in the morning, low in the afternoon, middle in the evening; and (7) Inconclusive type (“None of the above”).

The VJT [53] was designed by N. Marcoen and O. Mairesse to evaluate how sleepy survey participants thought they would be performing sleepiness-neutral activity (sitting and reading) at different randomly presented times after having habitual night sleep. Such sleep is expected to be terminated at approximately 7:30, either by a waking-up signal or due to a spontaneous awakening. The time cues from 8 a.m. to midday and from 8 p.m. to midnight are presented with one-hour intervals, while time cues between midday and 8 p.m. and after midnight are presented with two-hour intervals (Appendix A). Moreover, the participants would see a visual aid on the screen that consisted of clock times along the scale illustrating the daily variation in the outdoor illumination level and indicating the duration of the waking period (see [53] for these illustrations and other details). The randomly collected sleepiness self-ratings are then subsequently ordered for constructing a curve of expected sleepiness (Appendix A) for each survey participant and for averaging sleepiness within the morning, daytime, and evening/nighttime subintervals (Appendix A).

The SWAT [54,55] was constructed by A. Putilov for the assessment of the ability to sleep or wake at either appropriate or inappropriate sleep and waking times. The SWAT includes 10-item scales for assessing Morning Sleepability (MS), Nighttime Wakeability (NW), Daytime Wakeability (DW), and Daytime Sleepability (DS). A positive/negative score on each of these scales indicates ability/inability to sleep or wake on demand in the morning (MS; Appendix A), in the evening/at night (NW; Appendix A), and in the daytime (DW and DS; Appendix A).

All these questionnaire tools for chronobiological assessment were previously cross-validated (e.g., in [42,55]).

### 4.4. Statistical Analysis (Empirical Study)

We used the SPSS_26.0_ statistical software package (IBM, Armonk, NY, USA) for statistical analysis. Two-, three- and four-way ANOVAs of sleep times were run to test significance of the differences between the subdivisions (Appendix A) of the whole sample (Appendix A), and post hoc pairwise Bonferroni comparisons were then used to confirm significance of differences between subsamples of different chronotypes and/or weekday risetimes (Table 2 and Appendix A). Additionally, such ANOVAs were applied for comparison of morning chronotype with each of the other chronotypes (Table 3 and Appendix A). The differences between chronotypes are illustrated in Figure 4, Figure 5 and Appendix A. Finally, three-way ANOVAs of scores on questionnaire scales designed to assess sleepiness, sleepability, and wakeability in the morning, afternoon, and evening/night were used to confirm the differences between self-chosen chronotypes (Appendix A). The differences between chronotypes in these chronobiological characteristics are illustrated in Appendix A.

### 4.5. Model-Based Simulations (Simulation Study)

The least-squares method was used to simulate sleep times in morning and evening types (Figure 6 and Table 4). In the process of simulation, the empirical and simulated sleep times and the empirical and simulated weekday–weekend gaps in sleep times were compared several times until a drop in the difference between simulated and empirical value below 0.15 h (9 min) for each of these nine estimates (see the middle part of Table 4).

Generative artificial intelligence (GenAI) has not been used in this paper.

## 5. Conclusions

Either preferred or actual times for sleep and wakefulness are usually used to distinguish between morning and evening types. In contrast, such sleep times are not used for measurement of the circadian phase difference between these types, which is measured using conventional hormonal, physiological, and molecular rhythm markers of the circadian phase position. The literature suggests that the results of categorizations into chronotypes based on such measurement demonstrate low concordance with those based on sleep times. In the present in silico study, we suggested the possibility of using these sleep times for direct estimation of the differences between chronotypes in the circadian phase. The results suggested that the difference in circadian phase is equal to the difference in weekend risetime or bedtime when chronotypes are not significantly different in weekend–weekday gap in risetime and weekend time in bed. In the following empirical and simulation studies, we provided examples of the estimation of the circadian differences between several chronotypes. Overall, the results obtained underscore the possibility of using self-reported sleep times for estimation of the circadian phase differences between individuals in real-life situations, such as a 5-days-on/2-days-off school/work schedule. Additionally, we used the results of the comparison of weekday and weekend times in bed in different chronotypes to provide further evidence for our failure to sleep more on weekends to recover/compensate/pay back/catch up on lost sleep. The results of the comparison of various chronotypes corroborated the previously published results based on sleep times reported before and during lockdown, for early and later school start times, and for earlier and later weekday risers.

## Figures and Tables

**Figure 1 clockssleep-07-00067-f001:**
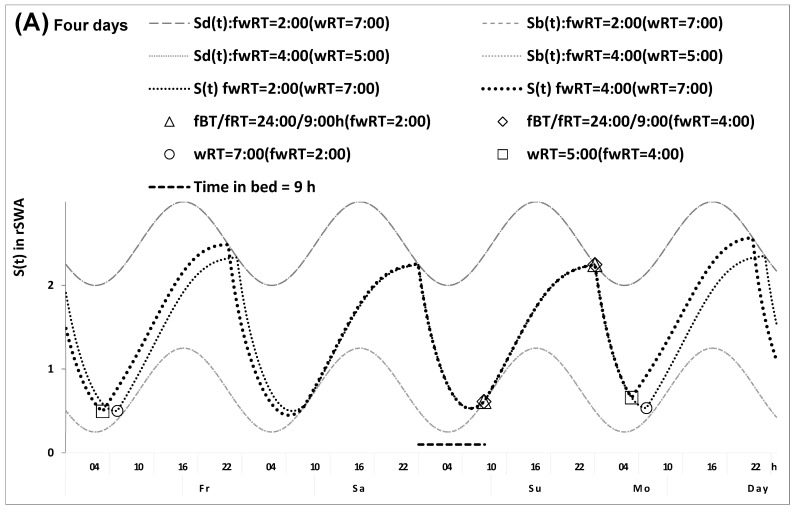
Sleep–wake cycles calculated for earlier and later risers with identical weekend risetimes. Sleep–wake cycles of earlier and later risers with identical weekend risetimes were calculated using a model of the sleep–wake regulating process. (**A**,**B**) Four- and one-day time courses (from Friday to Monday, Friday to Monday, and between Sunday and Monday, Sunday and Monday). *S_d_(t)* and *S_b_(t)*: The highest buildup and the lowest decay of *S(t)*, i.e., bedtime and risetime, respectively, determined by this endogenous sleep–wake regulating process *S(t)* (1,2) that is measured in rSWA (see the model parameters in Table 1). The risers differ in the weekend–weekday gap in risetime (fwRT, 2.0 h vs. 4.0 h) and have identical circadian phases of sleep, i.e., rise- and bedtimes on Sunday (fRT and fBT) are equal to 9:00 and 24:00, respectively. Therefore, they do not differ in weekend risetime, ΔfRT = 0. Due to the circadian modulation *C(t)* of the parameters of *S(t)*, the endogenously determined bed- and risetimes (shown for Sunday) are restored during just one day with ad lib sleep (between Friday and Saturday). The rate of this restoration is not affected by the extent of advancement relative to fRT. Consequently, fRT and fBT are expected to be identical after 5 days of earlier and later schedules with respect to (with respect to = 5.0 h and with respect to = 7.0 h, respectively), i.e., they are equal to 9:00 and 24:00 on Sunday. The results of model-based calculations suggest that, in the equation ΔwRT = ΔfRT + (−ΔfwRT), ΔfRT = 0 (ΔwRT = ΔfwRT). However, these calculations do not take into account that a larger shift relative to fRT (4.0 h vs. 2.0 h) leads to a larger advancing shift in light exposure, which in turn leads a greater advance of the circadian clock phase, and consequently a larger advance of the wake and sleep phases of the sleep–wake cycle on weekdays and even on weekends. Therefore, a non-zero difference in fRT (ΔfRT) is expected instead of the calculated zero difference (i.e., a larger advance of fRT after a larger fwRT). Consequently, it is expected that, in the equation ΔwRT = ΔfRT + (−ΔfwRT), ΔfRT ≠ 0, contrary to the suggested ΔfRT = 0.

**Figure 2 clockssleep-07-00067-f002:**
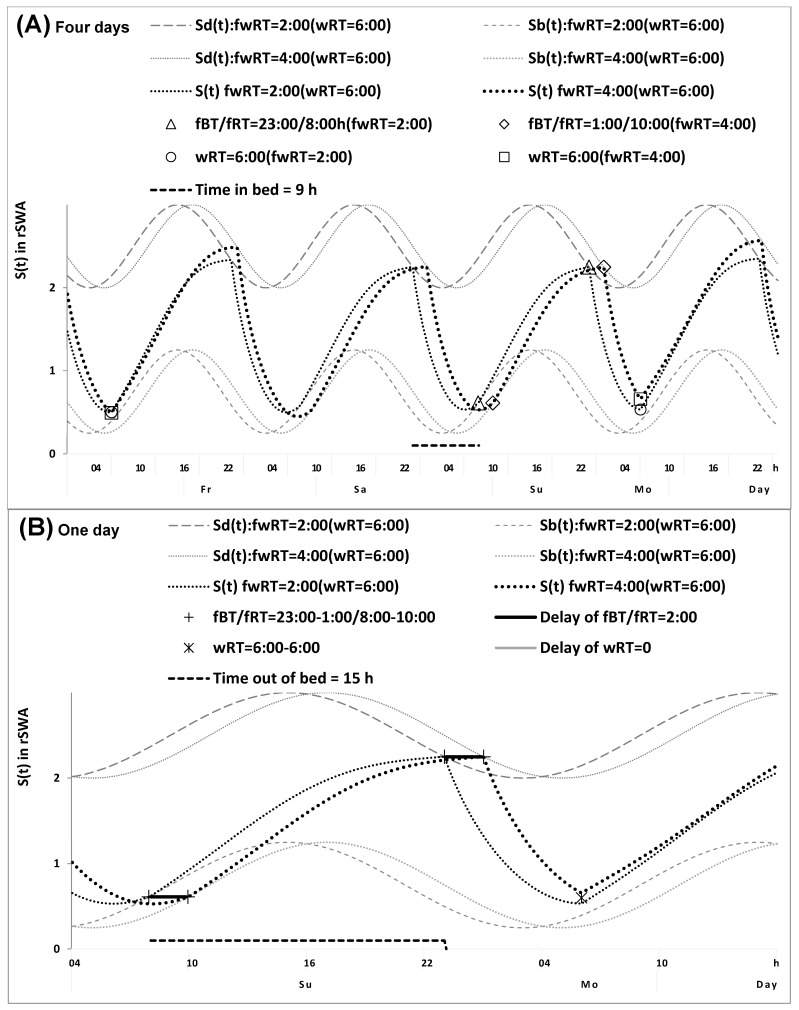
Sleep–wake cycles calculated for earlier and later risers with identical weekday risetimes. Sleep–wake cycles of earlier and later risers with two different fwRT values (2.0 h or 4.0 h, as in Figure 1), and, additionally, with a 2.0 h difference in the circadian phase of sleep that makes ‘with respect to’ identical in these risers. Due to the combination of the 2.0 h difference in fwRT with the 2.0 h difference in fBT/fRT, the difference ‘with respect to’ is equal to zero (ΔwRT = 0). Similarly to calculations in Figure 2, these calculations do not take into account that a larger shift relative to fRT (4.0 h vs. 2.0 h) leads to a larger advancing shift in light exposure, etc. (see Figure 1). Consequently, it is expected that, in the equation ΔwRT = ΔfRT + (−ΔfwRT), ΔfRT – ΔfwRT ≠ 0, contrary to the suggested ΔfRT − ΔfwRT = 0. (**A**) Four cycles and (**B**) One cycle.

**Figure 3 clockssleep-07-00067-f003:**
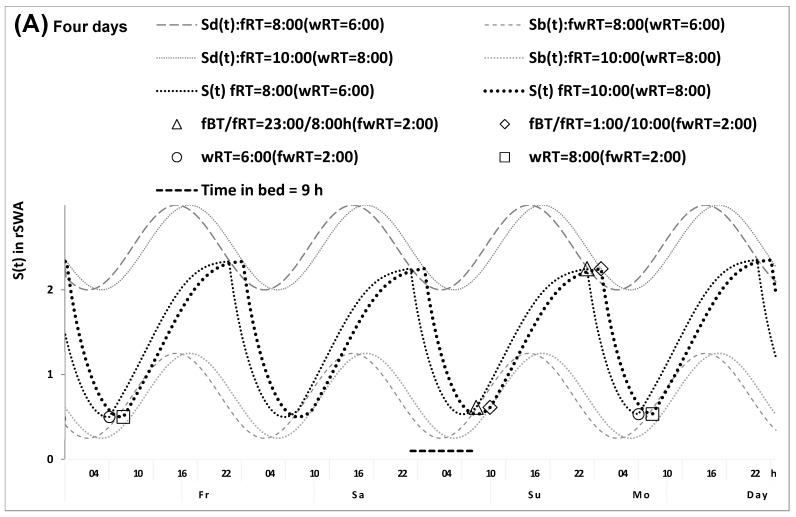
Sleep–wake cycles calculated for earlier and later risers with an identical weekend-weekday gap in risetime. Sleep–wake cycles of earlier and later risers with an identical (2.0 h) gap and the 2.0 h difference in the circadian phase of sleep, ΔfwRT = 0. When there is no difference in fwRT, the difference ‘with respect to’ is equal to the difference in fRT/fBT, ΔfRT = ΔwRT, ΔfwRT = ΔfRT − ΔwRT = 0. Consequently, ΔfRT is equal to the difference between earlier and later risers in the circadian phase of sleep, in contrast to the cases shown in Figure 1 and Figure 2, when the difference between the risers is shorter due to a larger advance of fRT in later than earlier risers caused by a larger advancing shift in light exposure (see Figure 1). This figure predicts the necessity for similarity in fwRT (ΔfRT − ΔwRT ≈ 0), between earlier and later risers for estimating the difference between them in the circadian phase (ΔfRT ≈ ΔwRT ≈ 2.0 h). (**A**) Four cycles and (**B**) One cycle.

**Figure 4 clockssleep-07-00067-f004:**
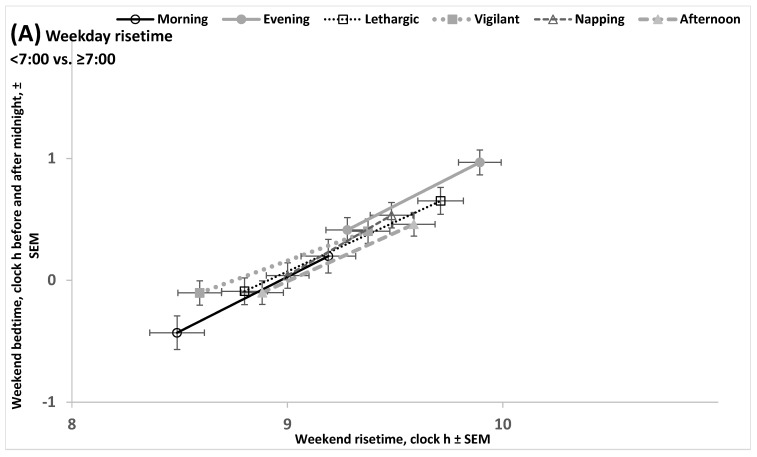
Shift in weekend bed- and risetimes associated with the difference in weekday risetime. (**A**) Weekend bed- and risetimes and their standard errors were calculated for earlier or later weekday risers (either earlier than 7:00 or later) of morning type and one of the other types (see these sleep times in Appendix A). (**B**) Weekend bed- and risetimes and their standard errors were calculated for each chronotype either without correction or with correction for the difference in weekday risetime (see Appendix A). This correction designated the earlier of two bed- or risetimes in morning types and the later of two bed- or risetimes in each of the other chronotypes. Since such a correction leads to similarity in weekend–weekday gaps in risetime, the difference in weekend risetime is an estimate of the difference between morning and another type in the circadian phase of sleep.

**Figure 5 clockssleep-07-00067-f005:**
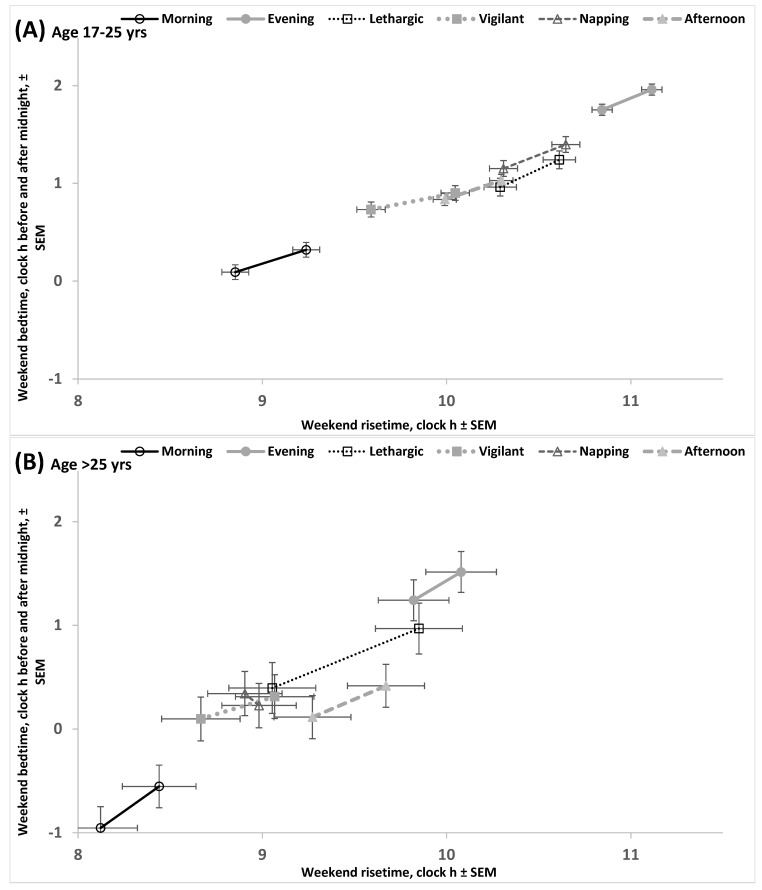
Shift in weekend bed- and risetimes associated with the difference in weekday risetime in different ages. (**A**,**B**) Participants of younger and older age, respectively. Weekend bed- and risetimes and their standard errors were calculated for each chronotype either without correction or with correction for the difference in weekday risetime (see Appendix A). This correction suggested the earlier of two bed- or risetimes in morning types and the later of two bed- or risetimes in each of the other chronotypes. Since such a correction leads to similarity of weekend–weekday gaps in risetime, the difference in weekend risetime is an estimate of the difference between morning and another type in the circadian phase of sleep. These estimates are similar for different ages despite the profound difference between ages in weekend sleep timing.

**Figure 6 clockssleep-07-00067-f006:**
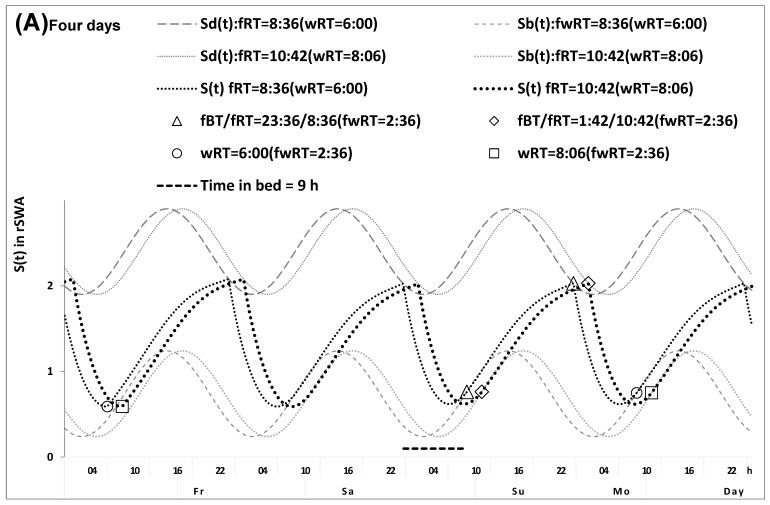
Simulation of sleep times reported by earlier and later weekday risers of morning and evening types. (**A**,**B**) Simulation of four and one sleep–wake cycle for confirmation of the model-based predictions of the in silico study (Figure 3A,B) and for confirmation of the result of the empirical study (Figure 4B). The result suggested the only difference between morning and evening types lies in the circadian phase of sleep. See Table 4 for the model parameters, simulated sleep times, and the difference between reported and simulated sleep times.

**Table 1 clockssleep-07-00067-t001:** Sleep times on weekdays and weekends are calculated for four different weekend–weekday gaps in risetime.

		Weekend-Weekday Gap in Risetime, h		4	3	2	1	2 − 4
Sleep times	Time in bed	From Tuesday to Wednesday (wTiB)		6.75	7.20	7.70	8.29	0.95
calculated		From Sunday to Monday(fRT)		5.00	6.00	7.00	8.00	2.00
for two	Risetime	Wednesday (with respect to)		5.00	6.00	7.00	8.00	2.00
days of		Sunday (fRT)		8.97	8.98	8.99	9.00	0.02
the week	Bedtime	Wednesday (wBT)		22.05	22.72	23.28	23.71	1.23
		Sunday (fBT)		24.00	24.00	24.00	24.00	0.00
Parameters	Circadian	Circadian phase, clock h	*φ_max_*	16.00	16.00	16.00	16.00	0.00
of the	parameters	Circadian amplitude, rSWA	*A*	0.50	0.50	0.50	0.50	0.00
model	(2)	Circadian period, h	*τ*	24.00	24.00	24.00	24.00	0.00
(1a, 1b, 2)	Homeostatic	Highest buildup, rSWA	*Sd*	2.50	2.50	2.50	2.50	0.00
	parameters	Lowest decay, rSWA	*Sb*	0.75	0.75	0.75	0.75	0.00
	(1a, 1b)	Upper asymptote, rSWA	*Su*	4.51	4.51	4.51	4.51	0.00
		Lower asymptote, rSWA	*Sl*	0.70	0.70	0.70	0.70	0.00
		Time constant for buildup, h	*Tb*	24.75	24.75	24.75	24.75	0.00
		Time constant for decay, h	*Td*	2.30	2.30	2.30	2.30	0.00
		Twofold circadian impact	*k*	2.00	2.00	2.00	2.00	0.00

Notes. Results of the in silico study. The identical parameters of the model (lower part) were used for the calculation of sleep–wake cycles with four weekday–weekend gaps in risetime (4, 3, 2, and 1 h). Since the calculations suggest that the duration of ad lib sleep at night between Saturday and Sunday is endogenously determined, the model predicts practically identical time in bed, risetime, and bedtime for these days after any of four weekday risetimes. Clock time is given in decimal hours; 2−4. Figure 1, Figure 2 and Figure 3 illustrate the identity of weekend sleep duration and timing after any shift in weekday risetime (either at 5:00 or 7:00 for obtaining a 4.0 h and 2.0 h gap, respectively).

**Table 2 clockssleep-07-00067-t002:** Difference between chronotypes in some of the sleep times.

Type	Morning	Evening	Lethargic	Vigilant	Napping	Afternoon
Type	Difference in time in bed on weekends					
Morning	-	−0.08	−0.30	0.14	−0.20	−0.25
Evening	0.94 ***	-	−0.22	0.22	−0.12	−0.17
Lethargic	0.42 ***	−0.52 ***	-	0.44 **	0.09	0.05
Vigilant	0.33 *	−0.61 ***	−0.09	-	−0.34 *	−0.39 **
Napping	0.76 ***	−0.19	0.33 *	0.42 **	-	−0.05
Afternoon	0.34 **	−0.61 ***	−0.09	0.00	−0.42 ***	-
	Difference in time in bed on weekdays					
	Difference in weekend–weekday gap in risetime					
Morning		−1.02 ***	−0.72 ***	−0.20	−0.96 ***	−0.58 ***
Evening	−1.22 ***		0.30	0.83 ***	0.06	0.44 ***
Lethargic	−0.89 ***	0.33 **		0.53 **	−0.24	0.14
Vigilant	−0.24	0.98 ***	0.65 ***		−0.77 ***	−0.39 *
Napping	−0.93 ***	0.29 **	−0.03	−0.68 ***		0.38 *
Afternoon	−0.72 ***	0.51 ***	0.18	−0.47 ***	0.21	
	Difference in weekend–weekday gap in time in bed					

Notes. Results of ANOVAs with the following post hoc pairwise comparisons of times in bed on weekends and weekdays (upper part, above and below the diagonal, respectively) and weekend–weekday gaps in risetime and times in bed (lower part, above and below the diagonal, respectively). Type: Distribution into chronotypes (lines) chosen among the options, illustrating and shortly describing the patterns of daily change in alertness level. Significance of the differences between types in sleep time with Bonferroni correction for the number of post hoc pairwise comparisons: * *p* < 0.05, ** *p* < 0.01, *** *p* < 0.001 for t-score. The results support the predictions of a non-significant difference in fTiB (ΔfTiB) between morning and evening types despite the significant difference between them in weekday time in bed wTiB (ΔfTiB). See also Appendix A for the distribution into chronotypes, Appendix A for the results of ANOVAs preceding these post hoc pairwise comparisons, and Appendix A (upper part) for sample-averaged sleep times obtained by averaging within the whole sample.

**Table 3 clockssleep-07-00067-t003:** Difference in weekend rise- and bedtime between morning types and each of the other chronotypes.

Another Type		Evening	Lethargic	Vigilant	Napping	Afternoon
Difference between Another and Morning type (uncorrected for risetime)						
Risetime	ΔfRT	1.49 ***	0.83 ***	0.29	0.81 ***	0.79 ***
Bedtime	ΔfBT	1.61 ***	0.80 ***	0.53 **	0.81 ***	0.59 ***
Difference between later and earlier weekday risers, ≥ 7 minus < 7						
Risetime	ΔfRT	0.62 ***	0.91 ***	0.78 ***	0.48 **	0.70 ***
Bedtime	ΔfBT	0.56 ***	0.74 ***	0.51 **	0.50 **	0.56 ***
Difference between Another ≥ 7 and Morning < 7 (corrected for risetime)						
Weekend	ΔfRT	2.11 ***	1.74 ***	1.07 ***	1.29 ***	1.50 ***
Weekend	ΔfBT	2.17 ***	1.54 ***	1.04 ***	1.30 ***	1.15 ***

Notes. Results of ANOVAs provided the estimates of differences between sleep times of Morning types and each of the other types (Appendix A): Evening, Lethargic, Vigilant, Napping, and Afternoon (Another). Upper part: Difference in sleep times between the Another type and the Morning type before correction for their difference in weekend–weekday gap in risetime (Another minus Morning). The middle part shows the difference between earlier and later weekday risers (either earlier than 7:00 or later, ≥7 minus < 7) in an analyzed sample consisting of the Morning type and one of the other types. The lower part shows the difference between earlier weekday risers of Morning type and later weekday risers of another type (Another ≥ 7 minus Morning < 7), i.e., the correction for the difference in the weekend–weekday gap in risetime that gives the difference between the circadian phases of sleep in this pair of (Morning and Another) types. Difference in the lower part can be directly calculated as the sum of the upper part and middle part differences: ≥7 − <7 + Another-Morning = (Morning ≥ 7 + Another ≥ 7 − Morning < 7 − Another < 7)/2 + (Another < 7 + Another ≥ 7 − Morning < 7 − Morning ≥ 7)/2 = Another ≥ 7 − Morning < 7. See Appendix A for the differences in other sleep times. Level of significance: ** *p* < 0.01, *** *p* < 0.001 for t-score.

**Table 4 clockssleep-07-00067-t004:** Simulation of sleep times reported by earlier morning and later evening type risers.

	Type			Morning	Evening
Simulated	Time	Weekday	wTiB	6.88	6.96
sleep	in	Weekend	fTiB	8.92	8.88
time, h	bed	Gap	fwTiB	2.04	1.92
or clock h	Rise-	Weekday	RT	5.99	8.10
	time	Weekend	fRT	8.49	10.61
		Gap	fwRT	2.50	2.51
	Bed-	Weekday	wBT	23.11	25.14
	time	Weekend	fBT	23.57	25.73
		Gap	fwBT	0.46	0.59
Difference	Time	Weekday	wTiB	0.07	−0.14
between	in	Weekend	fTiB	−0.06	0.01
reported	bed	Gap	fwTiB	−0.01	−0.01
and	Rise-	Weekday	RT	−0.02	−0.02
simulated	time	Weekend	fRT	−0.08	0.10
sleep		Gap	fwRT	0.04	−0.05
time, h	Bed-	Weekday	wBT	0.12	−0.15
	time	Weekend	fBT	−0.13	0.14
		Gap	fwBT	−0.01	−0.01
Parameters	Circadian	Circadian phase, clock h	*φ_max_*	14.40	16.50
of the	parameters	Circadian amplitude, rSWA	*A*	0.50	0.50
model	(2)	Circadian period, h	*τ*	24.00	24.00
(1a, 1b, 2)	Homeostatic	Highest buildup, rSWA	*Sd*	2.40	2.40
	parameters	Lowest decay, rSWA	*Sb*	0.74	0.74
	(1a, 1b)	Upper asymptote, rSWA	*Su*	5.20	5.20
		Lower asymptote, rSWA	*Sl*	0.70	0.70
		Time constant for buildup, h	*Tb*	39.40	39.40
		Time constant for decay, h	*Td*	2.57	2.57
		Twofold circadian impact	*k*	2.00	2.00

Notes. Results of the simulation study. Example of simulation of sleep times reported by morning types with weekday risetime < 7:00 and evening types with weekday risetime ≥ 7:00. The same set of model parameters with the exception of the circadian phase (*φ*: 14.40 vs. 16.50), were used for the morning and evening types, respectively. Reported sleep times were averaged within 442 morning types with weekday risetime < 7:00 and within 714 evening types with weekday risetime ≥ 7:00 (see Appendix A). Morning and evening in the middle part of the table: Reported sleep time minus simulated sleep time, h, for morning and evening types with weekday risetime < 7:00 and later, respectively. Clock time is given in decimal hours. See also notes to Table 1 and illustration of the results of this simulation in Figure 6.

## Data Availability

Data from the survey and simulation results are available from the first author on reasonable request.

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
