# Peer review of "Estimation of the Circadian Phase Difference in Weekend Sleep and Further Evidence for Our Failure to Sleep More on Weekends to Catch Up on Lost Sleep"

_2624-5175, 2025, doi:10.3390/clockssleep7040067_

Round 1

Reviewer 1 Report

Comments and Suggestions for Authors

Having performed an in silico study followed by empirical and simulation studies  based on the self-reports about sleep timing in a large cohort (over 4000 responders), the authors proved that self-reported sleep times can be used for the estimation of the circadian phase differences.

The aim is clearly stated. The methods are comprehensively described. The results are consistent and detailed. The figures and tables are well-done and provide important information which is not duplicated in the text.

There are few minor comments:

  • Could you please specify which software was used for the in silico study and modelling?
  • Line 73: Please correct the following: “…these rhythmicities that it known”. There should be “is” instead of “it”
  • The number of self-citations is 11 (out of 72 references) that might be a bit too many. Some of these citations are methodological and important for the current study. However, it would be reasonable to reconsider self-citations.
  •                                                                                                                                   

Author Response

Reply to Reviewer #1

Open Review

Quality of English Language

( ) The English could be improved to more clearly express the research.
(x) The English is fine and does not require any improvement.

Yes

Can be improved

Must be improved

Not applicable

Is the content succinctly described and contextualized with respect to previous and present theoretical background and empirical research (if applicable) on the topic?

(x)

( )

( )

( )

Are the research design, questions, hypotheses and methods clearly stated?

(x)

( )

( )

( )

Are the arguments and discussion of findings coherent, balanced and compelling?

(x)

( )

( )

( )

For empirical research, are the results clearly presented?

(x)

( )

( )

( )

Is the article adequately referenced?

(x)

( )

( )

( )

Are the conclusions thoroughly supported by the results presented in the article or referenced in secondary literature?

(x)

( )

( )

( )

Comments and Suggestions for Authors

Having performed an in silico study followed by empirical and simulation studies  based on the self-reports about sleep timing in a large cohort (over 4000 responders), the authors proved that self-reported sleep times can be used for the estimation of the circadian phase differences.

The aim is clearly stated. The methods are comprehensively described. The results are consistent and detailed. The figures and tables are well-done and provide important information which is not duplicated in the text.

Reply Summary. Thank you very much indeed for this evaluation, we agree with each of minor comments, and made appropriate changes in the manuscript. Please find the detailed responses below. The corresponding revisions/corrections were highlighted in the re-submitted files in red.

There are few minor comments:

  • Could you please specify which software was used for the in silico study and modelling?
  • Thank you for pointing this out. We added a sentence in the first subsection of the method section saying that all calculations and simulations were performed using macros in Excell software.
  • Line 73: Please correct the following: “…these rhythmicities that it known”. There should be “is” instead of “it”
  • Thank you for pointing this typo out. We corrected “it” (pun).
  • The number of self-citations is 11 (out of 72 references) that might be a bit too many. Some of these citations are methodological and important for the current study. However, it would be reasonable to reconsider self-citations.
  • Reply. Thank you for pointing this out. We agree with this comment. Our calculations suggested a bit more than 15% self-citation of the whole reference list. Therefore, we reduced the number of self-citations along with the reduction of the whole list of references (from 72 to 55). This gave less than 15% self-citations allowed by this publisher.

Reviewer 2 Report

Comments and Suggestions for Authors

In this manuscript, the authors attempted to integrate theoretical mathematical modeling (in silico study), a large-scale empirical investigation (N=4940), and confirmatory simulations. The large dataset, in particular, is a notable strength of this work.
The main finding reported is that the preconditions of the in silico study were met when comparing specific subgroups: "morning types with weekday risetime before 7:00" and "evening types with weekday risetime at or after 7:00." In this specific comparison, the difference in weekend risetime exceeded two hours, which the authors interpret as the true circadian phase difference. The simulation study reportedly succeeded in replicating these findings.

The authors conclude that it is possible to estimate circadian phase differences from self-reported sleep times under specific conditions and that their findings provide further evidence disproving the concept of weekend catch-up sleep.
However, the manuscript suffers from multiple fundamental flaws that severely undermine the validity of its conclusions.

# Major Issues

1.  Inability to Validate Estimation Accuracy: Lack of a Gold Standard
    Although the study's primary aim is the "estimation" of circadian phase difference, it completely lacks a comparison with a gold-standard objective measure (e.g., minimum core body temperature, Dim Light Melatonin Onset). This is a fatal methodological flaw.
    Without knowing how accurately the proposed method's output (the difference in weekend risetime) reflects the true biological circadian phase, it is impossible to scientifically evaluate the study's main results. The conclusion of a "more than 2.0-h difference" remains unverified against any objective physiological marker.
    The lack of validation means that the study's conclusions are unsubstantiated assertions.
    To claim the validity of the proposed method, validation of its estimation accuracy against an objective physiological marker is essential. If such data are unavailable, the study should be framed as a purely hypothetical modeling paper, and the conclusion that the method can "estimate" the phase difference must be retracted.

2.  Fundamental Flaw in Research Logic: Circular Reasoning
    The logical structure of this study is fundamentally flawed by circular reasoning (tautology). The authors do not test a hypothesis; rather, they search for a case within their data that fits their pre-defined conditions and then conclude that this confirms their theory.
    The in silico study concludes that for the proposed method to be valid, the difference in the weekend-weekday risetime gap (`ΔfwRT`) between the compared groups must be close to zero. The subsequent empirical study then intentionally searches for subgroups that satisfy this exact condition and performs the analysis only on that specific pair.
    This circular logic means that the generalizability of the proposed method is not demonstrated at all.
    The authors must explicitly acknowledge this circular logic in the "Limitations" section. To genuinely test their hypothesis, they should apply the model to the entire dataset and demonstrate how it can correct for a non-zero `ΔfwRT` to still yield a valid phase estimate.

3.  Lack of Validity and Logical Contradiction of the Chronotyping Method
    The study relies entirely on a novel "Single-Item Chronotyping" (SIC) tool, but not only is its scientific validity undemonstrated, its use creates a serious contradiction within the paper's overall logic.
    The scientific basis for the seven SIC types reflecting stable, distinct biological circadian characteristics is unclear. No rigorous, independent validation against objective biological markers or established questionnaires is presented.
    In the Introduction, the authors argue that chronotype is a direct reflection of the biological clock's circadian phase and that existing self-report questionnaires are imprecise. Despite this, their analysis uses the SIC, their own subjective tool that is not based on objective circadian phase, creating a direct contradiction between their stated premise and their actions. This lack of logical consistency severely damages the study's credibility.
    The authors must provide a compelling explanation for the scientific rationale of using the SIC and how it aligns with the logic presented in the Introduction. If this is not possible, the analysis should be entirely redone using a standard chronotype measure (e.g., MCTQ), and the SIC should be described merely as an experimental tool in the limitations.

4. Concerns Regarding the Validity of the Mathematical Model: Arbitrary Parameter Settings
    The parameter settings for the mathematical model used in this study are highly opaque and appear to be arbitrary.
    There is no explanation of the prior research or theoretical basis for the core parameters of the model, such as the homeostatic time constant and the bed/risetimes used in the simulation. The parameters shown in Table 4, in particular, were likely adjusted post-hoc to fit the observed data (as suggested by the phrase "slightly modified"), indicating a risk of overfitting rather than true predictive power.
    Conclusions derived from a model based on unsubstantiated parameters lack generalizability and scientific validity.
    The authors must provide a detailed description of the theoretical basis, cited literature, or decision process for all model parameters. If parameters were adjusted to fit the data, this procedure must be clarified, and the risk of overfitting and its impact on the interpretation of the results must be thoroughly discussed.

5.  Lack of Focus and Thematic Disunity in the Manuscript
    This paper attempts to address two weakly related themes simultaneously: "estimation of circadian phase difference" and "disproving weekend catch-up sleep." This dual focus blurs the paper's overall message and weakens the persuasiveness of each argument.
    The logical rationale connecting the two themes is poor, leading to reader confusion. The argument against "catch-up sleep," in particular, feels disconnected from the primary goal of phase estimation and appears abrupt.
    To clarify the paper's scientific contribution, I strongly recommend the sections related to "catch-up sleep" (parts of the Introduction, and the associated Results and Discussion) should be removed entirely to concentrate on improving the quality of the phase estimation argument.

6. Unclear Significance of "Group Difference" Estimation
    If an individual's circadian phase could be accurately estimated from their sleep patterns, it would be a highly useful, non-invasive tool. However, the method proposed in this study is limited to estimating the difference between "groups" that meet highly specific conditions.
    The academic or clinical significance of estimating a phase difference under such limited conditions is unclear.
    The authors must clearly articulate the applicability and scientific contribution of their method. Specifically, they should explain why estimating a "group difference" is important and what its significance is.

7.  Serious Concerns Regarding Statistical Analysis: Post-Hoc Subgroup Selection and Risk of Data Dredging
    The analytical strategy of this study is highly likely to constitute "data dredging"—a post-hoc exploration of data in search of a positive result, rather than a scientifically valid hypothesis test.
    It is suggested that an initial analysis comparing all "morning types" versus all "evening types" failed to meet the model's preconditions. Instead of reporting this primary result, the authors subdivided the sample by chronotype (7 categories) and weekday risetime (2 categories) and conducted numerous post-hoc pairwise comparisons.
    This approach dramatically inflates the risk of a Type I error (a false positive). By conducting multiple comparisons across numerous subgroups, the probability of finding a "statistically non-significant difference" by chance becomes very high.
    The authors must clearly state that this analysis is exploratory and post-hoc. The primary negative finding—that the model's preconditions were not met when comparing all morning vs. evening types—should be explicitly reported and discussed as a main result.

The followings are minor Issues

8. Excessive Use of Citations
    For an original research article, this manuscript cites an excessive number of references, including many that do not appear essential for the logical development of the arguments.
    For example, the paragraph on "catch-up sleep" in the Introduction lists 22 references (ref. 33-54), which serves more as a bibliography than a focused discussion.
    The authors should carefully select their citations, presenting only the most critical references that directly and strongly support each claim.

9.  Clarity and Organization of the Manuscript
    The manuscript is generally very dense and difficult to read. The logical connections between the *in silico*, empirical, and simulation studies are often unclear, and many sentences are excessively long and complex. Authors should improve clarity.

10. Interpretability of Figures and Tables
    Many of the figures and tables are not self-explanatory and are difficult to interpret. Figure 1, in particular, is hard to understand due to overlapping lines and a highly complex legend.
    All figures and tables should be redesigned to be simpler and more self-contained, so that the key information is highlighted. 

Author Response

Reply to Reviewer #2

Open Review

Quality of English Language

(x) The English could be improved to more clearly express the research.
( ) The English is fine and does not require any improvement.

Reply. The English was improved, may sentences were rewritten or corrected to more clearly express the study.

Yes

Can be improved

Must be improved

Not applicable

Is the content succinctly described and contextualized with respect to previous and present theoretical background and empirical research (if applicable) on the topic?

( )

( )

(x)

( )

Are the research design, questions, hypotheses and methods clearly stated?

( )

( )

(x)

( )

Are the arguments and discussion of findings coherent, balanced and compelling?

( )

( )

(x)

( )

For empirical research, are the results clearly presented?

( )

(x)

( )

( )

Is the article adequately referenced?

( )

( )

(x)

( )

Are the conclusions thoroughly supported by the results presented in the article or referenced in secondary literature?

( )

( )

(x)

( )

Comments and Suggestions for Authors

In this manuscript, the authors attempted to integrate theoretical mathematical modeling (in silico study), a large-scale empirical investigation (N=4940), and confirmatory simulations. The large dataset, in particular, is a notable strength of this work.
The main finding reported is that the preconditions of the in silico study were met when comparing specific subgroups: "morning types with weekday risetime before 7:00" and "evening types with weekday risetime at or after 7:00." In this specific comparison, the difference in weekend risetime exceeded two hours, which the authors interpret as the true circadian phase difference. The simulation study reportedly succeeded in replicating these findings.

The authors conclude that it is possible to estimate circadian phase differences from self-reported sleep times under specific conditions and that their findings provide further evidence disproving the concept of weekend catch-up sleep.
However, the manuscript suffers from multiple fundamental flaws that severely undermine the validity of its conclusions.

Reply Summary. Many thanks for this evaluation. We made many appropriate changes in the manuscript. Please find the detailed responses below. The corresponding revisions/corrections were highlighted in the re-submitted files in red.

# Major Issues

  1.  Inability to Validate Estimation Accuracy: Lack of a Gold Standard
      Although the study's primary aim is the "estimation" of circadian phase difference, it completely lacks a comparison with a gold-standard objective measure (e.g., minimum core body temperature, Dim Light Melatonin Onset). This is a fatal methodological flaw.
      Without knowing how accurately the proposed method's output (the difference in weekend risetime) reflects the true biological circadian phase, it is impossible to scientifically evaluate the study's main results. The conclusion of a "more than 2.0-h difference" remains unverified against any objective physiological marker.
        The lack of validation means that the study's conclusions are unsubstantiated assertions.
        To claim the validity of the proposed method, validation of its estimation accuracy against an objective physiological marker is essential. If such data are unavailable, the study should be framed as a purely hypothetical modeling paper, and the conclusion that the method can "estimate" the phase difference must be retracted.

Reply 1. Thank you for pointing this out. We agree with this comment in that the proposed method was not validated using parallel measurements of the difference in circadian phase of the rhythms-markers of the biological clock. However, we cannot say that “the study's primary aim is the "estimation" of circadian phase difference”. We said everywhere that this aim was fop arose a methodology or to show the possibility of “estimation”. As Comment #5 says “If an individual's circadian phase could be accurately estimated from their sleep patterns, it would be a highly useful, non-invasive tool”. Therefore, we do not think it is not unimportant purpose of our study to suggest such a tool and exemplify it application to analysis of a large dataset rather than to additionally validate it in this particular way. Consequently, we used the division into chronotypes for exemplifying this method because there were many previous publications that all agreed that there is a 2-h or a bit larger than 2-h difference between morning and evening types in the circadian phase of melatonin, core body temperature and some other rhythms-markers of the circadian clock. Our results on direct estimation of the difference in the circadian phase of sleep were in full agreement with these previous findings. In two paragraphs of Discussion (including the last, limitation paragraph) we pointed at this validation insufficiency of our study and at the prospects of further studies aimed on validation of the proposed methodology with the parallel measurements of the rhythms-markers of the circadian phase. In the revised version of the manuscript, we extended these sentences to more clearly express these shortcomings and prospects. We, however, expect the problems with such validation because, as we pointed out in the manuscript, there exists a difference between chronotypes as well as between ages in the phase angle between sleep timing and the circadian phase of a rhythm-marker of the biological clock.

  1.  Fundamental Flaw in Research Logic: Circular Reasoning
      The logical structure of this study is fundamentally flawed by circular reasoning (tautology). The authors do not test a hypothesis; rather, they search for a case within their data that fits their pre-defined conditions and then conclude that this confirms their theory.
      The in silico study concludes that for the proposed method to be valid, the difference in the weekend-weekday risetime gap (`ΔfwRT`) between the compared groups must be close to zero. The subsequent empirical study then intentionally searches for subgroups that satisfy this exact condition and performs the analysis only on that specific pair.
        This circular logic means that the generalizability of the proposed method is not demonstrated at all.
        The authors must explicitly acknowledge this circular logic in the "Limitations" section. To genuinely test their hypothesis, they should apply the model to the entire dataset and demonstrate how it can correct for a non-zero `ΔfwRT` to still yield a valid phase estimate.

Reply 2. Thank you for pointing this out. We suggested an additional analysis of this dataset to demonstrate that it might not be necessary to “performs the analysis only on that specific pair”. On the results of this additional analysis, we replaced Table 3 by a new Table and new supplementary tables (S14-S16) that show the result of the analysis of the whole dataset of morning and another chronotype (including evening type: see also the calculation in the note to Tables 3 and S16). These new approach to estimation and the results of its application indicated that it is not necessary to limit the data analysis to “…intentionally searches for subgroups that satisfy this exact condition and performs the analysis only on that specific pair”. Thus, these new results of this additional analysis suggested that there is a way of using the whole dataset consisting from morning type and one of another types for estimation of the difference between this pair of chronotypes in the circadian phase of sleep. See also other Replies #6 and #7.

  1.  Lack of Validity and Logical Contradiction of the Chronotyping Method
      The study relies entirely on a novel "Single-Item Chronotyping" (SIC) tool, but not only is its scientific validity undemonstrated, its use creates a serious contradiction within the paper's overall logic.
      The scientific basis for the seven SIC types reflecting stable, distinct biological circadian characteristics is unclear. No rigorous, independent validation against objective biological markers or established questionnaires is presented.
        In the Introduction, the authors argue that chronotype is a direct reflection of the biological clock's circadian phase and that existing self-report questionnaires are imprecise. Despite this, their analysis uses the SIC, their own subjective tool that is not based on objective circadian phase, creating a direct contradiction between their stated premise and their actions. This lack of logical consistency severely damages the study's credibility.
        The authors must provide a compelling explanation for the scientific rationale of using the SIC and how it aligns with the logic presented in the Introduction. If this is not possible, the analysis should be entirely redone using a standard chronotype measure (e.g., MCTQ), and the SIC should be described merely as an experimental tool in the limitations.

Reply 3. Thank you for pointing this out. The last (19th) question of the MEQ asks to self-classify an individual as one of four possible classes, definitely morning type (Self-MM), more morning than evening type (Self-M), more evening than morning type (Self-E), definitely evening type (Self-EE). Several studies found that this single question version of the MEQ is a stronger correlate of objective (e.g., including heredity) measures of chronotype. The SIC can be viewed as a further development of such self-classification but based on multi-dimensional approach to chronotyping. First. We discussed this approach in one of the paragraphs of Discussion. Second. We used multi-item questionnaires to demonstrate validity of the classification based on the SIC (Figures S4-S7, especially Figure S4 showing the daily variation in sleepiness in these chronotypes). Third. We did not discuss this in more details to escape from further increase of complexity of this paper, but more details can be found in our previous publications (e.g., including one paper published in the last issue of Clocks@Sleep: “Owls are not what they seem…”). Forth. In the revised version of the manuscript, we more clearly express, in a separate subsection of Results that the significant differences between these chronotypes were confirmed using such chronobiological characteristics as sleep times and sleepiness, sleepability, and wekeability in the morning, daytime and evening/night hours (Tables S2, S17 and Figures S1-S7). Therefore, our study does not “rely entirely on a novel "Single-Item Chronotyping" (SIC) tool”. Rather, the results on the SIC are supported by the reported sleep times (they are in the focus of this study) and multi-item multiscale assessments of time course of sleepiness and sleepability, and wekeability in the morning, daytime and evening/night.

  1. Concerns Regarding the Validity of the Mathematical Model: Arbitrary Parameter Settings
      The parameter settings for the mathematical model used in this study are highly opaque and appear to be arbitrary.
      There is no explanation of the prior research or theoretical basis for the core parameters of the model, such as the homeostatic time constant and the bed/risetimes used in the simulation. The parameters shown in Table 4, in particular, were likely adjusted post-hoc to fit the observed data (as suggested by the phrase "slightly modified"), indicating a risk of overfitting rather than true predictive power.
        Conclusions derived from a model based on unsubstantiated parameters lack generalizability and scientific validity.
        The authors must provide a detailed description of the theoretical basis, cited literature, or decision process for all model parameters. If parameters were adjusted to fit the data, this procedure must be clarified, and the risk of overfitting and its impact on the interpretation of the results must be thoroughly discussed.

Reply 4. Thank you for pointing this out. To respond to this concern, we extended Method section by adding a detailed description of the way of applying this model (1,2) for particular calculations of the differences between sleep cycles in Figures 1-3. Since further enlargement of the description of the model and parameters might make the text of our paper even more complicated, we also added one new sentence saying that the detailed description of the model, its physical counterpart, and its parameters can be found in one of the cited papers (Front Netw Physiol. 2023, 3:1285658: “Reaction of the endogenous regulatory mechanisms to early weekday wakeups…”). The description of the derivation of the initial parameters of the model is given in brief in the first subsection of the method section. And…, finally, we are wondering how it can happen that “the parameters shown in Table 4, in particular, were likely adjusted post-hoc to fit the observed data”? First, we consulted the date of the excel files with calculations used for Figures 1-3 and Table 1. These files were saved on May 21st, 2018 (our guess is that this was the time we started to work on our paper in Frontiers in Physiology. 2018, 9, 1529: “Simulation of the ontogeny of social jet lag…”). Second. The date of saving the excel file with the simulation performed for Figure 6 and Table 4 is September 3rd, 2025. Third. For the calculations saved on May 21st, 2018 and now used in Figures 1-3 and Table 1, we suggested that time in bed for free days is exactly 9.00 h with rise- and bedtimes set exactly at 9:00 and 24:00, respectively. The parameters for these, highly arbitrarily chosen rounded sleep times (!) derived on May 21st, 2018 (!), were only “slightly (!) modified” to simulate empirical data on September 3rd, 2025 (!). Looks like we, instead, made the accurate prediction of the empirical results of the present study 7 yrs ago by taking such highly arbitrarily rounded sleep times for deriving the model parameters for the self-reported rise- and bedtimes (“just took this number from the ceiling”, as this Russian idiom says!).

  1.  Lack of Focus and Thematic Disunity in the Manuscript
      This paper attempts to address two weakly related themes simultaneously: "estimation of circadian phase difference" and "disproving weekend catch-up sleep." This dual focus blurs the paper's overall message and weakens the persuasiveness of each argument.
      The logical rationale connecting the two themes is poor, leading to reader confusion. The argument against "catch-up sleep," in particular, feels disconnected from the primary goal of phase estimation and appears abrupt.
        To clarify the paper's scientific contribution, I strongly recommend the sections related to "catch-up sleep" (parts of the Introduction, and the associated Results and Discussion) should be removed entirely to concentrate on improving the quality of the phase estimation argument.

Reply 5. Thank you for pointing this out. We modified the description of results of the in silico study and some other paragraphs of the manuscript to stress more clearly that these two predictions of the model are closely related. The prediction 1 illustrated by Figure 1 says that weekday sleep duration cannot be increased due to a shorter weekday sleep duration (thus "disproving weekend catch-up sleep."). It follows that weekend sleep timing can be used for estimation of difference between circadian phase of sleep in chronotypes characterized in the real-life conditions by different weekday sleep duration that is the prediction 2 illustrated in Figures 2 and 3 (the possibility of "estimation of circadian phase difference" as the difference in weekend sleep timing).

  1. Unclear Significance of "Group Difference" Estimation
      If an individual's circadian phase could be accurately estimated from their sleep patterns, it would be a highly useful, non-invasive tool. However, the method proposed in this study is limited to estimating the difference between "groups" that meet highly specific conditions.
      The academic or clinical significance of estimating a phase difference under such limited conditions is unclear.
        The authors must clearly articulate the applicability and scientific contribution of their method. Specifically, they should explain why estimating a "group difference" is important and what its significance is.

Reply 6. Thank you for pointing this out, especially, for stressing that “If an individual's circadian phase could be accurately estimated from their sleep patterns, it would be a highly useful, non-invasive tool.” In order to respond to the following “However…” (“…the method proposed in this study is limited to estimating the difference between "groups" that meet highly specific conditions.   The academic or clinical significance of estimating a phase difference under such limited conditions is unclear”, we suggested another way of estimation of this difference using the whole set of data on two compared chronotypes (Tables 3 and S14, S16, and see also Replies 2 and 7, as well the note to Tables 3 and S16, illustrated by Figure 4A and 4B). Moreover, the previously described methodology can be, in fact, applied to compare circadian phases of sleep in any couple of individuals with similar weekend-weekday gap in risetime. We did not emphasize this possibility in this manuscript 1) due to intension to avoid its further complication and 2) due to the absence of information supporting this particular application of the proposed methodology (in contrast, the application of this methodology to chronotypes was supported by numerous publications where the morning and evening types were compared using the rhythms-markers of the circadian phase and always the same ≥2 h difference in the circadian phase was reported).

  1.  Serious Concerns Regarding Statistical Analysis: Post-Hoc Subgroup Selection and Risk of Data Dredging
      The analytical strategy of this study is highly likely to constitute "data dredging"—a post-hoc exploration of data in search of a positive result, rather than a scientifically valid hypothesis test.
      It is suggested that an initial analysis comparing all "morning types" versus all "evening types" failed to meet the model's preconditions. Instead of reporting this primary result, the authors subdivided the sample by chronotype (7 categories) and weekday risetime (2 categories) and conducted numerous post-hoc pairwise comparisons.
        This approach dramatically inflates the risk of a Type I error (a false positive). By conducting multiple comparisons across numerous subgroups, the probability of finding a "statistically non-significant difference" by chance becomes very high.
        The authors must clearly state that this analysis is exploratory and post-hoc. The primary negative finding—that the model's preconditions were not met when comparing all morning vs. evening types—should be explicitly reported and discussed as a main result.

Reply 7. Thank you for pointing this out. Indeed, the subsection of Methods about statistical analysis in the previous version of the manuscript was too short to avoid the misunderstanding of the analysis from which this post hoc comparison came (it was however explained in the notes to many tables). In response, we extended this subsection to say more clearer that this was ANOVA with the following post-hoc pairwise comparison of chronotypes (the comparison, of cause, was made with correction for the number of pairwise comparisons, as the table notes and the modified version of this subsection of Methods say). Definitely, there might be no any difference in terms of statistical pawer between the direct comparison of the factors in ANOVA (e.g., the main effect of factor “Chronotype”) and the post hoc pairwise comparison of the members of this factor (e.g., pairs of chronotypes). Besides, we think need not follow the recommendation (“The primary negative finding—that the model's preconditions were not met when comparing all morning vs. evening types—should be explicitly reported and discussed as a main result”), because, as already mentioned in Replies 2 and 6, we suggested another methodology that allows the comparison based on the whole sample of data on a couple of chronotypes, and this comparison is made by estimation of the effects of factors rather than by the following post hoc pairwise comparison because there are only two chronotypes in this analysis (Tables 3, S14-16, and Figures 4B and 5).

The followings are minor Issues

  1. Excessive Use of Citations
      For an original research article, this manuscript cites an excessive number of references, including many that do not appear essential for the logical development of the arguments.
      For example, the paragraph on "catch-up sleep" in the Introduction lists 22 references (ref. 33-54), which serves more as a bibliography than a focused discussion.
        The authors should carefully select their citations, presenting only the most critical references that directly and strongly support each claim.

Reply 8. Thank you for pointing this out. We agree with this comment, and we reduced the number of references (from 72 to 55) including the vast majority of references with “KEtchup sleep” in title (in fact, we can cite many dozen of such titles). We even deuced the number of citations of our own publications by leaving only those describing the cited methodologies and models.

  1.  Clarity and Organization of the Manuscript
      The manuscript is generally very dense and difficult to read. The logical connections between the *in silico*, empirical, and simulation studies are often unclear, and many sentences are excessively long and complex. Authors should improve clarity.

Reply 9. Thank you for pointing this out. We modified several sentences and enlarged several subsections of Results to stress the relationship between the results of the in silico, empirical, and simulation studies. Moreover, we added several clarifying sentences in entire text to additionally explain what a particular result implies for next set of results.

  1. Interpretability of Figures and Tables
      Many of the figures and tables are not self-explanatory and are difficult to interpret. Figure 1, in particular, is hard to understand due to overlapping lines and a highly complex legend.
      All figures and tables should be redesigned to be simpler and more self-contained, so that the key information is highlighted. 

Reply 10. Thank you for pointing this out. The figures and tables were redesigned to made them simpler or simply moved from the entire text of the manuscript to Supplementary Material (in it revised version it consists of a larger number of tables and figures). For instance, the previous Figures 1 and 2 were enlarged and the new Figures 1-3 have the same design as the enlarged Figure 6. Each of three legends to Figures 1-3 were shortened, partly due to moving to Methods those parts of the legends that explain the model and its parameters. The previous Figures 3-6 were moved to Supplementary Materials, while the previous Figures 7A-D and 8A-D were excluded, and, instead, much simpler and enlarged Figures 4AB and 5AB were added to illustrate a new Table 3 (the previous Table 3 was moved to Supplementary Materials to become Table S13), Table 1 was reduced, and so on, and so on…
